# Antigen glycosylation regulates efficacy of CAR T cells targeting CD19

Amanda Heard [1,10], Jack H. Landmann[1,10], Ava R. Hansen [2], Alkmini Papadopolou[3], Yu-Sung Hsu[1], Mehmet Emrah Selli [1], John M. Warrington [1], John Lattin[1], Jufang Chang[1], Helen Ha[1], Martina Haug-Kroeper[3], Balraj Doray [4], Saar Gill[5], Marco Ruella [5], Katharina E. Hayer[6,7], Matthew D. Weitzman [7,8], Abby M. Green[2,9], Regina Fluhrer [3] & Nathan Singh [1✉]

While chimeric antigen receptor (CAR) T cells targeting CD19 can cure a subset of patients with B cell malignancies, most patients treated will not achieve durable remission. Identification of the mechanisms leading to failure is essential to broadening the efficacy of this promising platform. Several studies have demonstrated that disruption of CD19 genes and transcripts can lead to disease relapse after initial response; however, few other tumor-intrinsic drivers of CAR T cell failure have been reported. Here we identify expression of the Golgi-resident intramembrane protease Signal peptide peptidase-like 3 (SPPL3) in malignant B cells as a potent regulator of resistance to CAR therapy. Loss of SPPL3 results in hyper-glycosylation of CD19, an alteration that directly inhibits CAR T cell effector function and suppresses anti-tumor cytotoxicity. Alternatively, over-expression of SPPL3 drives loss of CD19 protein, also enabling resistance. In this pre-clinical model these findings identify post-translational modification of CD19 as a mechanism of antigen escape from CAR T cell therapy.

[1] Division of Oncology, Washington University School of Medicine, Saint Louis, MO, USA. [2] Department of Pediatrics, Washington University School of Medicine, Saint Louis, MO, USA. [3] Biochemistry and Molecular Biology, Institute of Theoretical Medicine, Medical Faculty, University of Augsburg, Augsburg, Germany. [4] Division of Hematology, Washington University School of Medicine, Saint Louis, MO, USA. [5] Division of Hematology and Oncology, University of Pennsylvania School of Medicine, Philadelphia, PA, USA. [6] Department of Biomedical and Health Informatics, The Children's Hospital of Philadelphia, Philadelphia, PA, USA. [7] Department of Pathology and Laboratory Medicine, The Children's Hospital of Philadelphia, Philadelphia, PA, USA. [8] Department of Pathology and Laboratory Medicine, Perelman School of Medicine, University of Pennsylvania, Philadelphia, PA, USA. [9] Center for Genome Integrity, Siteman Cancer Center, Washington University School of Medicine, Saint Louis, MO, USA. [10]These authors contributed equally: Amanda Heard, Jack H. Landmann. ✉email: nathan.singh@wustl.edu

T cells engineered with chimeric antigen receptors (CARs) targeting the transmembrane protein CD19 have varied success in the treatment of B cell cancers. Response rates in pediatric patients with acute lymphoblastic leukemia (ALL) are high, with >85% achieving complete remission within 1 month of treatment[1,2]. Unfortunately, many patients with ALL who achieve remission ultimately relapse. Outcomes for patients with non-Hodgkin lymphoma or chronic lymphocytic leukemia (CLL) are more modest, with overall response rates of 30–50%[3–8]. These clinical data demonstrate that while curative for some, most patients treated with CAR T cells will not achieve long-term remission.

Failure of CAR T cells can result from tumor-intrinsic mechanisms, T cell-intrinsic mechanisms, or a combination of both. Several studies have identified T cell-intrinsic features that correlate with therapeutic failure, primarily related to memory differentiation status or expression of exhaustion-associated genes[9–12]. We previously reported that defects in cancer cell apoptotic signaling enable resistance to CAR T cell cytotoxicity that then drives the development of T cell dysfunction, implicating both cell types in disease progression[13]. Other studies have identified mechanisms by which modulation of CD19 surface expression can lead to resistance and relapse[14]. This process, broadly referred to as antigen escape, manifests as an apparent loss of surface CD19 expression by leukemic cells, making CAR T cells "blind" to their presence and permitting disease outgrowth. Antigen escape can occur at the genetic level via loss of entire CD19 alleles, resulting in failed protein expression, or loss of partial genes, resulting in expression of a truncated protein without the CAR binding epitope[15]. Alternative splicing of transcribed CD19 messenger RNA can eliminate the domains necessary for membrane integration or the CAR binding epitope, again resulting in loss of surface expression or loss of regions needed for CAR:antigen engagement[16]. To our knowledge, alterations of full-length CD19 protein that lead to resistance have not been reported.

Following translation, proteins can undergo several modifications. Among the most common post-translational modifications of secreted and membrane proteins is the addition of glycan residues as they transit to the cell surface. Glycosylation is an iterative process that is regulated by glycan-modifying enzymes found in the endoplasmic reticulum (ER) and Golgi apparatus and serves an essential role in protein folding, stability, and function. Signal peptide peptidase-like 3 (SPPL3) is an intramembrane aspartyl protease located in the Golgi that cleaves, among other targets, enzymes responsible for adding glycan residues to transiting transmembrane proteins[17,18]. This cleavage results in the release of the catalytically-active ectodomain of these glycosyltransferases, inhibiting them from adding glycans to proteins passing through the Golgi. Thus, SPPL3 functionally serves to restrict protein glycosylation. Antibodies and their derivative single-chain variable fragments (scFvs) that comprise CAR antigen-binding domains are variably sensitive to protein glycosylation for epitope recognition[19], and thus alteration of target antigen glycosylation presents a potential mechanism of escape from CAR T cell recognition.

We performed a genome-wide, CRISPR/Cas9-based loss-of-function screen in the human ALL cell line Nalm6 to identify genes whose function may promote resistance to CD19-targeted CAR T cell (CART19) cytotoxicity[13]. Here we show that disruption of SPPL3 caused CD19 hyperglycosylation which impairs the binding of anti-CD19 antibodies and impairs CAR T cell activation, thus enabling resistance to CAR therapy. We identify that hyperglycosylation of an asparagine residue proximal to the CAR binding epitope is directly responsible for enabling resistance. In contrast, overexpression of SPPL3 results in hypoglycosylation of CD19 which is followed by protein loss, profoundly impairing CAR T cell anti-leukemic efficacy. Our findings highlight the relevance of protein glycosylation in antigen:receptor interactions and identify post-translational modifications as essential regulators of CD19-targeted CAR T cell efficacy.

## Results

**Loss of SPPL3 results in resistance to CART19.** We engineered the human B-ALL cell line Nalm6 with the Brunello genome-wide guide RNA library[13,20] to enable the loss of function of a single gene within each Nalm6 cell. Following engineering, cells were combined with either control T cells or CD19-targeted CAR T cells (CART19). Co-cultures were collected 24 h later and underwent next-generation sequencing to identify which guide RNAs had been enriched in the surviving CART19-exposed cells, reflecting that loss of gene function promoted resistance to CART19 killing (Fig. 1a). We previously reported that the most enriched gene was FADD, encoding a membrane-proximal protein that initiates the apoptotic cascade upon binding of death receptors to their ligands[13]. Analysis of this same dataset revealed that the second-most enriched gene was SPPL3 (Fig. 1b), encoding a Golgi-resident protease[17,18]. To validate SPPL3's role in resistance to CART19, we disrupted genomic SPPL3 in Nalm6 using de novo designed guide RNAs targeting either exon 4 or 6 and generated clones that had biallelic disruption (clone 1 and 2, respectively). We confirmed that these clones lacked SPPL3 protein expression and that loss of SPPL3 did not enable an intrinsic growth advantage in Nalm6 cells (Supplementary Fig. 1a, b). To directly assess the impact of SPPL3 loss on CART19 cytotoxic efficacy, we established co-cultures of either SPPL3 wild-type (WT) or SPPL3^KO Nalm6 with CART19 and monitored Nalm6 survival over time. These studies confirmed that SPPL3^KO Nalm6 were resistant to CAR T cell killing both early during co-culture (48 h, Supplementary Fig. 1c) and over time (Fig. 1c). To validate that resistance was not unique to ALL, we disrupted SPPL3 in the CD19 + diffuse large B cell lymphoma line OCI-Ly10. As with Nalm6, SPPL3^KO OCI-Ly10 cells were also resistant to CART19 killing (Fig. 1d and Supplementary Fig. 1d), demonstrating that loss of SPPL3 enabled resistance in two distinct histologies of CD19 + malignancy. We extended these studies to our well-established xenograft model of systemic human ALL. NOD/SCID/γ^{−/−} (NSG) mice were engrafted with Nalm6 cells, either wild-type or SPPL3^KO, and treated with CART19 cells 6 days later. Consistent with our in vitro studies, loss of SPPL3 permitted leukemic progression in vivo (Fig. 1e and Supplementary Fig. 1e), which significantly shortened animal survival (Fig. 1f). Collectively, these data demonstrate that loss of SPPL3 enables robust resistance to CD19-targeted CAR therapy.

**Loss of SPPL3 in ALL impairs CART19 activation.** To determine if this resistance resulted from extrinsic impairments in T cell function or intrinsic resistance to apoptosis, we combined CART19 with either WT or SPPL3^KO Nalm6 and analyzed the expression of various surface markers of T cell activation (Fig. 2a). As compared to cells exposed to WT Nalm6, CART19 exposed to SPPL3^KO Nalm6 expressed lower levels of CD69, PD1, and TIM3 following overnight co-culture (Fig. 2b–d). Expression of CD107a (LAMP-1), a surrogate marker of lymphocyte degranulation, was also lower in CART19 exposed to SPPL3^KO Nalm6 (Fig. 2e). We established co-cultures combining WT or SPPL3^KO Nalm6 with immortalized T cells (Jurkat) engineered to express a CD19 CAR (Fig. 2a). These Jurkat cells were engineered with a fluorescent reporter system that transcribes GFP or eCFP upon activation of the central T cell transcription factors NFAT or

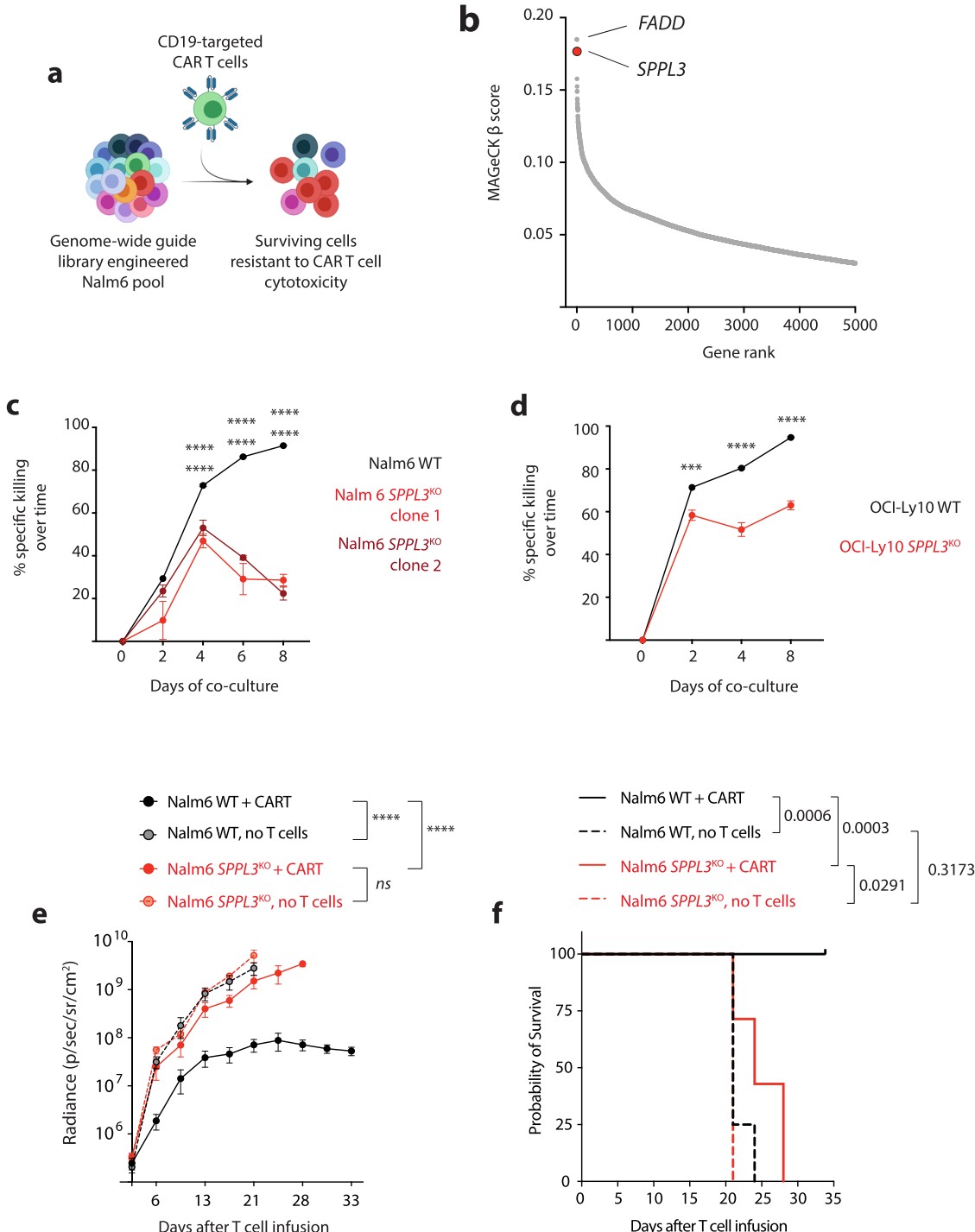

**Fig. 1 Loss of SPPL3 results in resistance to CD19-directed CAR T cells. a** Schematic of the genome-wide knockout screen in Nalm6 ALL. **b** Plot of MAGeCK beta scores for genes that were enriched in CART19 targeted cells compared to untargeted T cells. **c** Survival of Nalm6 and **d** OCI-Ly10 over time after combination with CART19 (E:T ratio 0.25:1). **e** Tumor radiance over time and **f** Survival of NSG mice treated engrafted with either WT or $SPPL3^{KO}$ Nalm6 after treatment with CART19 cells ($n = 7$ per group) or no treatment ($n = 4$ per group). ***$P < 0.001$, **** Error bars reflect mean ± standard error of the mean (s.e.m.). $P < 0.0001$ by two-way ANOVA with Bonferroni correction for multiple comparisons. Survival analyses were performed with the log-rank test. Data were representative of $n = 5$ (**c**, **d**) individual experiments with distinct donor T cells. In **c** statistical annotations reflect differences between WT and $SPPL3^{KO}$ clone 1 (upper) and WT and $SPPL3^{KO}$ clone 2 (lower). Source Data are provided as a Source Data file.

NFκB, respectively. Consistent with our findings in primary T cells, CAR19 Jurkat cells exposed to $SPPL3^{KO}$ Nalm6 demonstrated less transcription factor activation (Fig. 2f, g).

While signifying a quantitative decrease in receptor-driven stimulation, these findings did not demonstrate a complete lack of CAR-driven activation upon exposure to $SPPL3^{KO}$ Nalm6, as

indicated by the increase in activation marker expression compared to CAR-negative cells. We previously identified that partial resistance to CAR T cell cytotoxicity enables survival of some target cells, which leads to persistence of antigen. This persistence drives the development of T cell dysfunction, leading to a two-phased mechanism of therapeutic failure despite only

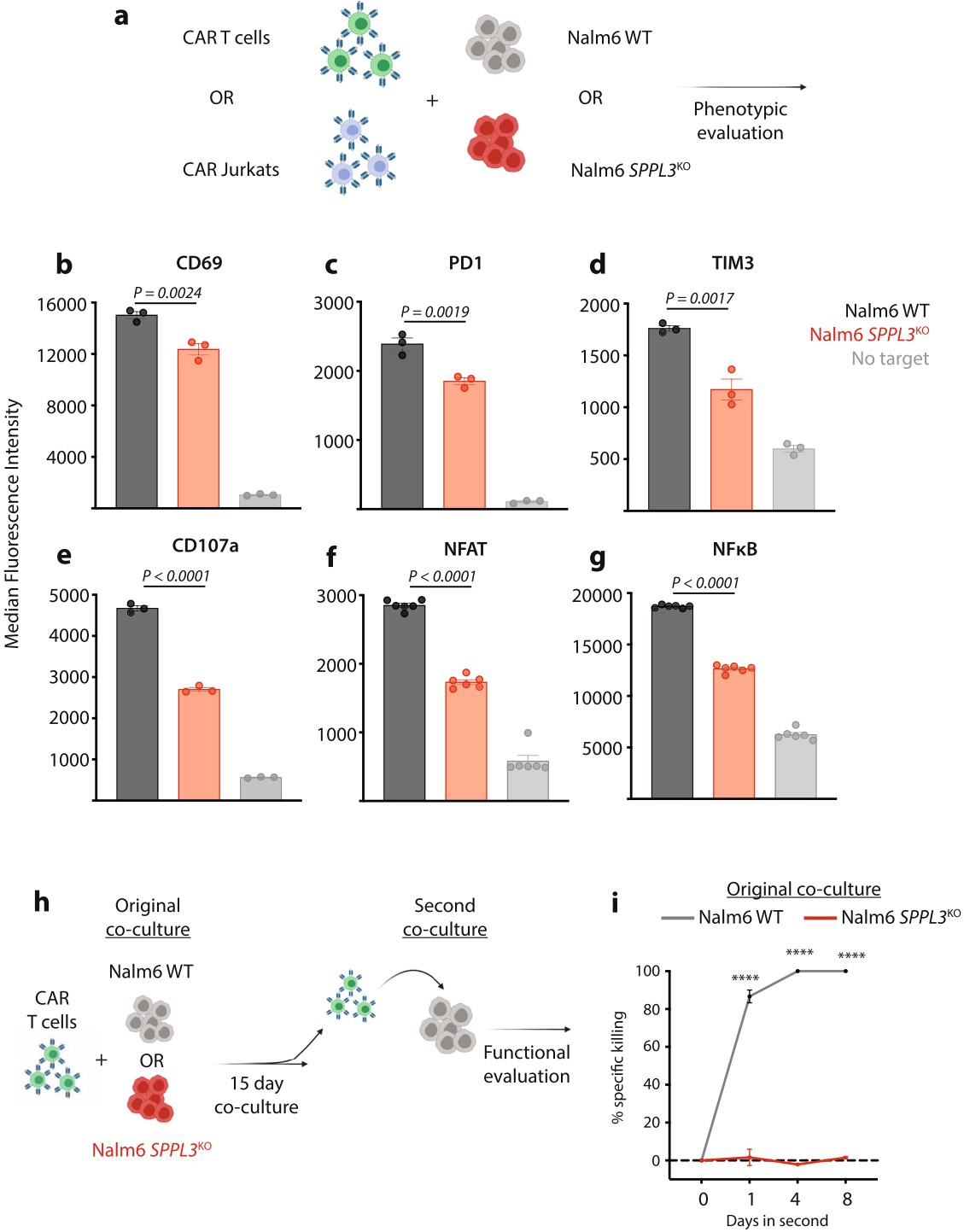

**Fig. 2 *SPPL3*^KO suppresses CD19 CAR T cell activation. a** Schematic of T cell activation studies. **b**–**d**, Expression of **b** CD69, **c** PD1, and **d** Tim3 on T cells after overnight co-culture with WT or *SPPL3*^KO Nalm6 cells or alone (control). **e** expression of CD107a on T cells after a 4-h co-culture with WT or *SPPL3*^KO Nalm6 cells. **f**, **g** Expression of **f** GFP and **g** eCFP, reflecting binding of NFAT and NFκB to their respective promoter sites in CD19 CAR-expressing Jurkat cells. **h** Schematic of long-term co-culture and re-challenge study. **i** Survival of WT Nalm6 over time after the combination of CART19 cells previously cultured with either WT or *SPPL3*^KO Nalm6 cells. **b**–**e**, **i** Representative of $n = 2$ individual experiments with distinct donor T cells. **f**, **g** representative of $n = 4$ individual experiments. Error bars reflect mean ± standard error of the mean (s.e.m.). ****$P < 0.0001$ by two-way ANOVA with Bonferroni correction for multiple comparisons. Source Data are provided as a Source Data file.

partial resistance to cytotoxicity[13]. To evaluate if this mechanism contributed to the resistance observed with *SPPL3* loss, we established two parallel cultures of CART19 with either WT or *SPPL3*^KO Nalm6 (Fig. 2h). These cultures were maintained for

15 days, at which time no residual WT Nalm6 remained but large quantities of *SPPL3*^KO Nalm6 persisted. We then isolated CART19 cells from these cultures and re-exposed them to WT Nalm6 to evaluate the functional capacity of these T cells.

CART19 cells originally exposed to WT Nalm6 rapidly cleared re-challenge while CART19 cells originally exposed to *SPPL3*[KO] Nalm6 were profoundly dysfunctional, with a complete inability to kill WT Nalm6 upon re-challenge (Fig. 2i). We observed the same trend when CARs contained the CD28 costimulatory domain (Supplementary Fig. 2a), confirming that this was a costimulation-independent effect. Together, these data demonstrate that CART19 cells undergo less antigen-driven activation by *SPPL3*[KO] Nalm6 and become progressively dysfunctional as a result of leukemic persistence, suggesting that *SPPL3*[KO] Nalm6 resistance may result from a combination of partial evasion of T cell killing followed by acquired T cell dysfunction.

**Hyperglycosylation of CD19 leads to CAR T cell failure.** While the function of SPPL3 has not been comprehensively defined, recent studies identified one of its primary roles as a Golgi-resident intramembrane aspartyl protease is to cleave enzymes that add glycans to extracellular asparagine residues, referred to as *N*-glycosylation[17,18]. The extracellular domain of CD19 contains seven asparaginse residues, five of which are glycosylated under normal conditions[21]. Analysis of protein lysates from both Nalm6 and OCI-Ly10 cells revealed that CD19 had an higher apparent higher molecular weight in the setting of SPPL3 loss (Fig. 3a and Supplementary Fig. 2b). To determine if changes in glycosylation were responsible for this increase in CD19 molecular weight, we treated Nalm6 lysates with two distinct glycosylases, peptide:*N*-glycosidase F (PNG) or Endo-glycosylase H (Endo H). These enzymes have distinct enzymatic activity: Endo H only cleaves core (mannose) residues added in the endoplasmic reticulum while PNG also cleaves branched glycans added in the Golgi. This approach allowed us to simultaneously determine if the increase in molecular weight was a result of *N*-glycosylation and the complexity of that glycosylation, a reflection of the organellar origin. PNG treatment resulted in a significant reduction in CD19 size in both WT and *SPPL3*[KO], while Endo H had minimal impact on CD19 from either cell type (Fig. 3b), suggesting the relative increase in molecular weight was a result of glycosylation that occurred in the Golgi, the site of SPPL3 activity. To identify the nature of altered CD19 glycosylation, we immunoprecipitated CD19 from WT and *SPPL3*[KO] Nalm6 and performed lectin blots. Staining of immunoprecipitated CD19 with concanavalin A, which binds mannose-rich glycan moieties, demonstrated no change with loss of SPPL3. Alternatively, staining with phytohaemagglutinin-L (PHA-L), which binds standard branched glycans, demonstrated a reduction in binding when SPPL3 was disrupted (Supplementary Fig. 2c). In conjunction with our glycosidase studies, this suggests that loss of SPPL3 results in both hyperglycosylation and structural alteration of branched glycans on CD19.

Given that antibodies and scFvs can be sensitive to glycoprotein structure, we hypothesized that hyperglycosylation may impair the CAR binding of CD19. To evaluate this, we stained Nalm6 cells with the same anti-CD19 antibody used to construct our CAR antigen-binding domain (clone FMC63). Notably, this is the same clone used to construct all FDA-approved CD19-targeted CAR products. At high antibody concentrations (1:100 – 1:400 dilution) there was no difference in FMC63 binding to CD19 on WT or *SPPL3*[KO] Nalm6 (Supplementary Fig. 3a), however, at lower concentrations (1:800–1:3200 dilution), there was a notable reduction in detection of CD19 on *SPPL3*[KO] Nalm6 (Fig. 3c, Supplementary Fig. 3b). We also interrogated the ability of a distinct CD19-directed antibody (clone HIB19) to recognize CD19 and observed less binding at all antibody concentrations on *SPPL3*[KO] Nalm6 (Supplementary Fig. 3c), suggesting that FMC63 is not uniquely sensitive to changes in glycosylation of CD19. We then treated WT and *SPPL3*[KO] Nalm6 with kifunensine, an inhibitor of mannosidase I that impairs glycoprotein processing,

for 10 days and repeated antibody staining. This treatment resulted in a reduction in CD19 apparent molecular weight, consistent with an impairment of protein glycosylation early in the E.R. (Fig. 3d), which rescued FMC63 binding on *SPPL3*[KO] Nalm6 cells (Supplementary Fig. 3d). We then repeated our in vitro cytotoxicity assay and found that inhibiting glycosylation with kifunensine also rescued CART19 efficacy against *SPPL3*[KO] Nalm6 cells, returning sensitivity to the levels seen for WT Nalm6 (Fig. 3e). These data directly implicate the altered glycan structure of CD19 as the etiology of resistance in the setting of SPPL3 loss.

To assess the impact of altered glycosylation on a distinct modality of CAR therapy, we turned to another B cell antigen that is the target of CAR therapy. Similar to CD19, CD22 is broadly expressed by B-lineage cells and is a well-described CAR target in ALL[22–24]. In contrast to CD19, deletion of *SPPL3* did not change CD22 electrophoretic mobility, suggesting no change in its glycosylation status (Fig. 3f). *SPPL3*[KO] also did not impact anti-CD22 antibody binding on Nalm6 cells (Supplementary Fig. 3e). Consistent with these observations observation, in vitro co-cultures demonstrated no difference in CD22-directed CAR T cell cytotoxicity against *SPPL3*[KO] and WT Nalm6, confirming that loss of SPPL3 did not enable resistance to CART22 (Fig. 3g and Supplementary Fig. 3f). This observation further supports that hyperglycosylation of CD19 is specifically responsible for resistance to CART19.

**Hyperglycosylation of CD19 N125 is responsible for failed CART19 function.** To define how increased N-glycosylation of CD19 resulted in failed CAR T cell function, we interrogated the role of specific CD19 asparagine residues. A previous study identified that CD19 residues W140, R144, and P203 are essential for FMC63 binding[21]. Modeling of the FMC63 binding epitope revealed two proximal asparagine residues, one of which is normally N-glycosylated (N125) and one of which is not (N114, Fig. 4a)[21]. Predicted intramolecular distance measurements reveal that N125 is closer to the binding epitope than N114, leading us to hypothesize that N125 is responsible for the disruption of the FMC63 binding epitope. To evaluate this, we disrupted *CD19* in our *SPPL3*[KO] Nalm6 cells to create a dual knockout (KO) cell line and introduced transgenic *CD19* constructs in which N114 or N125 had been mutated to glutamine, preventing glycosylation[21]. These constructs were encoded on a lentiviral expression vector that also encoded a truncated CD34 selection marker, enabling the purification of engineered cells independently of CD19 antibody binding. Using flow cytometry-assisted cell sorting, we produced cell lines that expressed equivalent amounts of CD19 (based on CD34 intensity) to directly evaluate how these mutations impacted anti-CD19 antibody binding capability. We found that introduction of WT CD19 into the dual KO cells resulted in CD19 antibody binding comparable to *SPPL3*[KO] Nalm6 (Fig. 4b), validating the sensitivity of this system. Dual KO cells engineered to express CD19[N114Q] demonstrated similar antibody binding as *SPPL3*[KO] Nalm6 and Dual KO_CD19[WT], indicating that preventing glycosylation of N114 had no impact on antibody binding. Dual KO cells engineered to express CD19[N125Q], however, demonstrated a significant increase in FMC63 binding, comparable to WT Nalm6 (Fig. 4b), indicating that preventing hyperglycosylation at N125 enhanced FMC63 binding despite SPPL3 loss. To confirm that this residue was not only responsible for rescuing antibody binding but also sensitivity to CAR cytotoxicity, we co-cultured these cell lines with CART19 cells. Consistent with antibody binding, dual KO_CD19[N125Q] cells were as sensitive to CART19 as WT Nalm6 (Fig. 4c). These data demonstrate that loss of SPPL3 prevents CART19 activity by hyperglycosylating CD19 at N125.

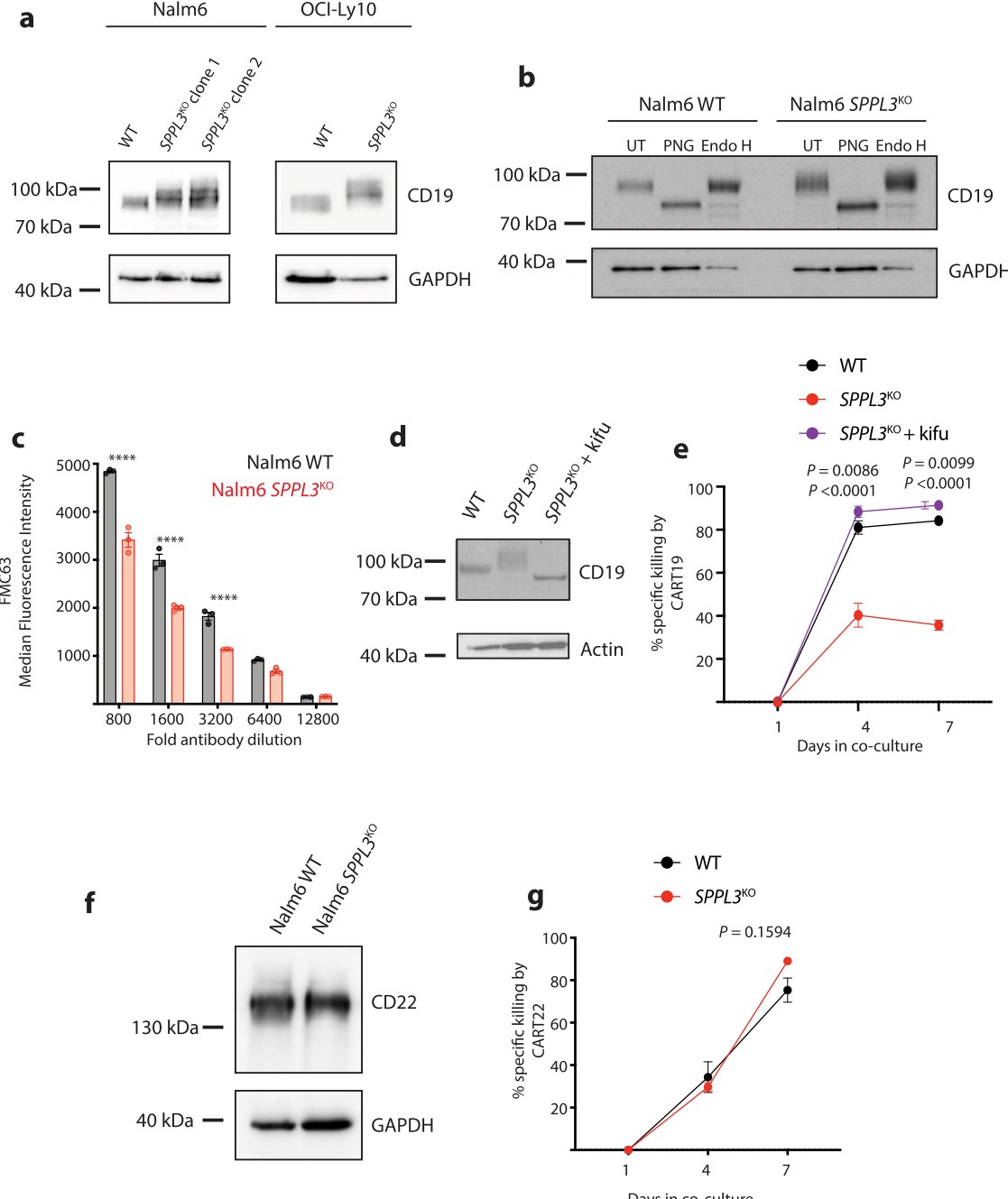

**Fig. 3 Hyperglycosylation impairs detection of CD19. a** Western blot of lysates from WT or *SPPL3*KO Nalm6 and OCI-Ly10 cells probed for CD19. Protein electrophoresis was performed on a 6% polyacrylamide gel. Representative of $n = 4$ individual experiments. **b** Western blot of lysates from WT or *SPPL3*KO Nalm6 cells that were either untreated (UT), treated with PNGase F (PNG), or treated with Endoglycosidase H (Endo H) and then probed for CD19. Representative data of $n = 3$ individual experiments. Electrophoresis performed on a 6% polyacrylamide gel. **c** Median Fluorescence Intensity of CD19 as detected by FMC63-APC on the surface of WT or *SPPL3*KO Nalm6 cells. **d** Western blot analysis of lysates from cells treated with kifunensine (16 ng/mL) for 10 days and then probed for CD19. Representative of $n = 2$ individual experiments. **e** Survival of WT, untreated *SPPL3*KO, and kifunensine-treated *SPPL3*KO Nalm6 over time in co-culture with CART19 cells (E:T ratio 0.25:1) statistical annotations reflect differences between *SPPL3*KO + kifu and WT and *SPPL3*KO + kfiu and *SPPL3*KO (lower). **f** Western blot of lysates from WT or *SPPL3*KO Nalm6 cells probed for CD22. Representative of $n = 4$ individual experiments. **g** Survival of WT or *SPPL3*KO Nalm6 cells over time after combination with CART22 (E:T ratio 0.25:1). **e, g** Representative data from $n = 3$ individual experiments with distinct donor T cells. Error bars reflect mean ± standard error of the mean (s.e.m.). *$P < 0.05$, **$P < 0.01$, ***$P < 0.001$, ****$P < 0.0001$ by two-way ANOVA with Bonferroni correction for multiple comparisons. Source Data are provided as a Source Data file.

**Increased SPPL3 activity results in loss of surface CD19.**
Overexpression of SPPL3 has been shown to cause protein hypoglycosylation via increased glycosyltransferase cleavage[17]. To investigate how hypoglycosylation of CD19 impacted CART19 efficacy, we over-expressed SPPL3 in *SPPL3*KO Nalm6 (*SPPL3*KO+) using a

lentiviral expression vector that again contained a CD34 selection marker. We purified CD34 + cells 5 days after engineering and found that overexpression of SPPL3 resulted in a reduction of CD19 molecular weight (Supplementary Fig. 4a), similarly to what we observed with kifunensine treatment, and consistent with

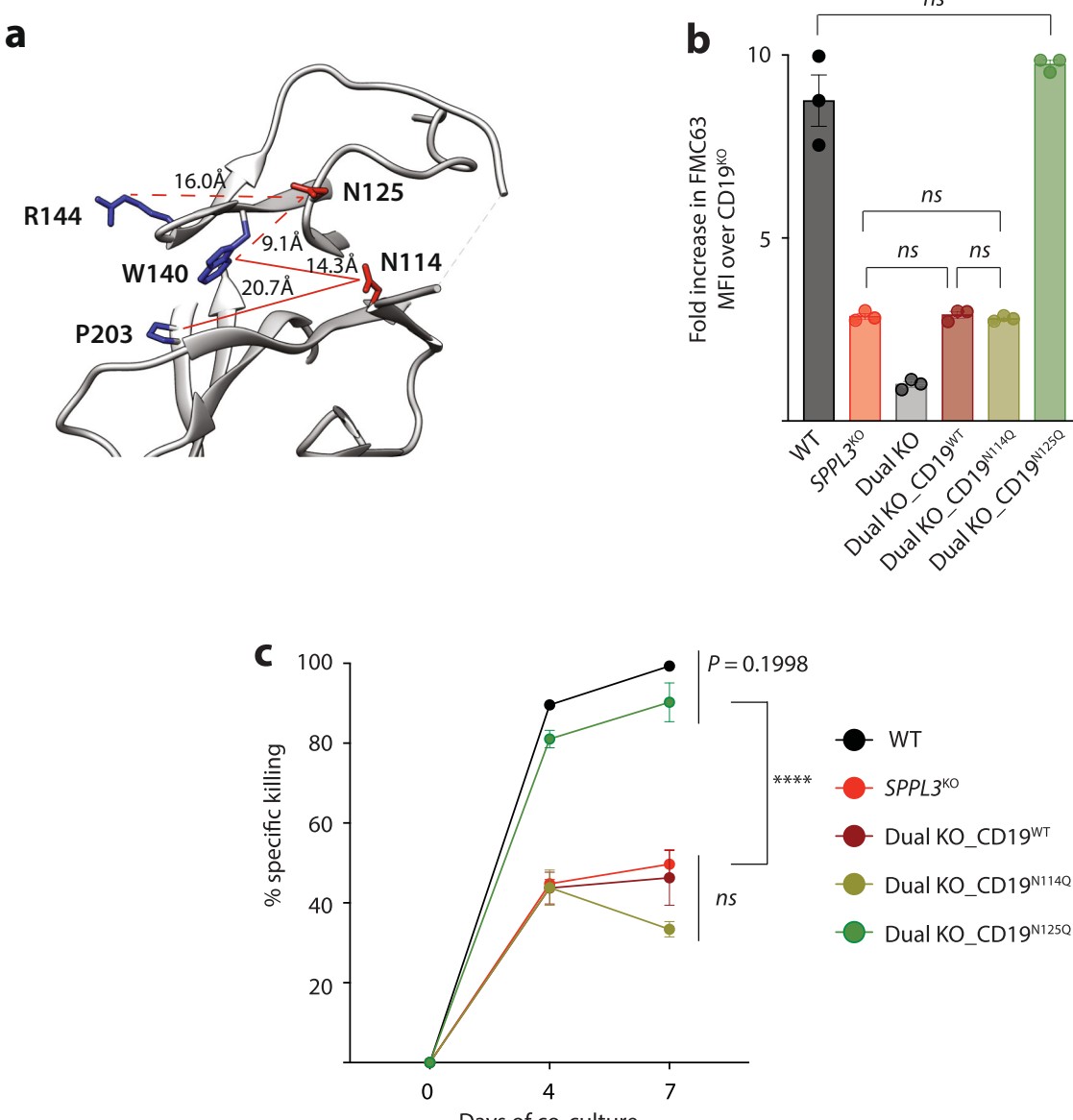

**Fig. 4 Hyperglycosylation of asparagine 125 is responsible for impairing CART19 efficacy. a** Structural modeling of the FMC63 binding epitope on CD19. Residues in blue (R144, W140, and P203) are essential for antibody binding and proximal asparagine residues are in red (N125 and N114). **b** Median Fluorescence Intensity of CD19 as detected by FMC63-APC on the surface of WT, $SPPL3^{KO}$ or $SPPL3^{KO}$ $CD19^{KO}$ (dual KO) Nalm6 cells engineered to express mutated CD19 molecules. Representative data from $n = 2$ flow cytometry experiments. **c** Survival of Nalm6 cells over time after combination with CART19 (E:T ratio 0.25:1). Representative data from $n = 2$ experiments with independent T cell donors. Error bars reflect mean ± standard error of the mean (s.e.m.). ****$P < 0.0001$ by two-way ANOVA with Bonferroni correction for multiple comparisons. Source Data are provided as a Source Data file.

hypoglycosylation. Evaluation of these cells over time revealed a progressive impairment in FMC63 binding to CD19 on $SPPL3^{KO}$ + cells after engineering, with the emergence of a clear CD19-negative population that slowly increased in proportion (Fig. 5a). Staining with the HIB19 anti-CD19 antibody clone revealed a similar shift in binding with SPPL3 overexpression (Supplementary Fig. 4b). We hypothesized that this loss of surface expression was due to intracellular retention of the hypoglycosylated variant of CD19. To evaluate this, we performed a fractionated cellular lysis to isolate cytosolic and membrane compartments. We found that, as opposed to lysates collected early after engineering (Supplementary Fig. 4a), lysates collected several weeks later were devoid of CD19 in all compartments (Fig. 5b). Quantitative reverse-transcriptase PCR confirmed that SPPL3 over-expressing cells were still transcribing the *CD19* gene (Supplementary Fig. 4c), suggesting post-transcriptional

loss of CD19. Consistent with this progressive loss, $SPPL3^{KO}$ + Nalm6 became increasingly resistant to CART19. $SPPL3^{KO}$ + Nalm6 combined with CART19 10 days after engineering demonstrated modest resistance (Supplementary Fig. 4d). In contrast, co-cultures established 20 days after the re-introduction of SPPL3 revealed profound resistance with minimal CART19 function (Fig. 5c). These findings indicate that hypoglycosylation of CD19 also leads to antigen escape as a result of protein loss.

## Discussion
Understanding mechanisms of resistance to CAR T cells is fundamental to improving the efficacy of this platform in both hematologic and solid malignancies. Beyond enhancing therapeutic activity, identification of tumor-intrinsic features that lead to resistance will allow for more appropriate patient selection,

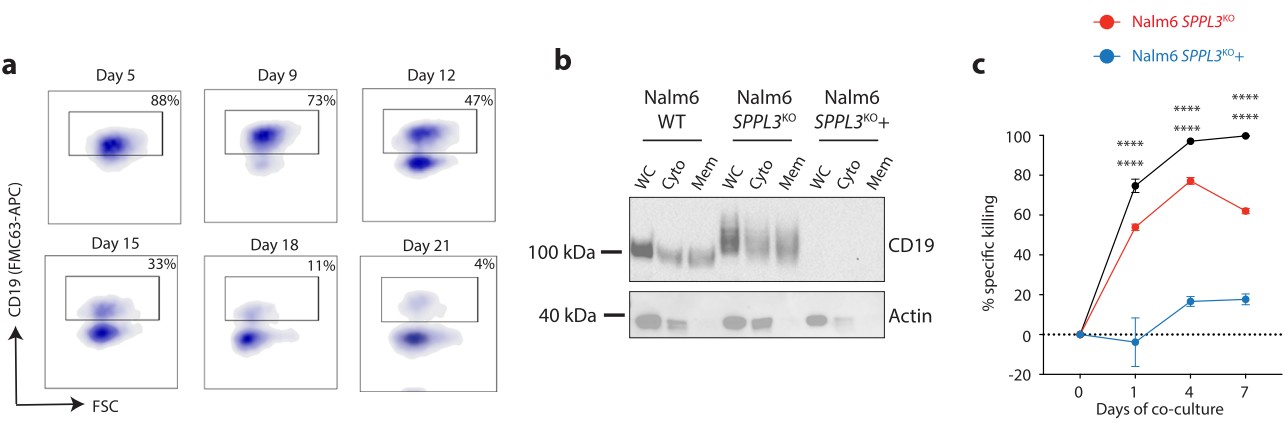

**Fig. 5 Overexpression of SPPL3 results in loss of CD19 surface expression. a** Sequential flow cytometric evaluation of FMC63-APC binding on *SPPL3*KO + Nalm6 after overexpression of SPPL3. **b** Western blot of fractionated cellular lysates from WT, *SPPL3*KO or *SPPL3*KO with overexpression of SPPL3 (*SPPL3*KO+) Nalm6 cells probed for CD19 performed 20 days after engineering. Representative of n = 2 individual experiments. **c** Survival of WT, *SPPL3*KO, or *SPPL3*KO + Nalm6 over time when established on 20 days after re-introduction of SPPL3 (E:T 0.25:1). Representative data from n = 2 experiments with independent T cell donors. Error bars reflect mean ± standard error of the mean (s.e.m.). ****P < 0.0001 by two-way ANOVA with Bonferroni correction for multiple comparisons. Statistical annotations reflect differences between WT and *SPPL3*KO (upper) or WT and *SPPL3*KO + Nalm6 (lower). Source Data are provided as a Source Data file.

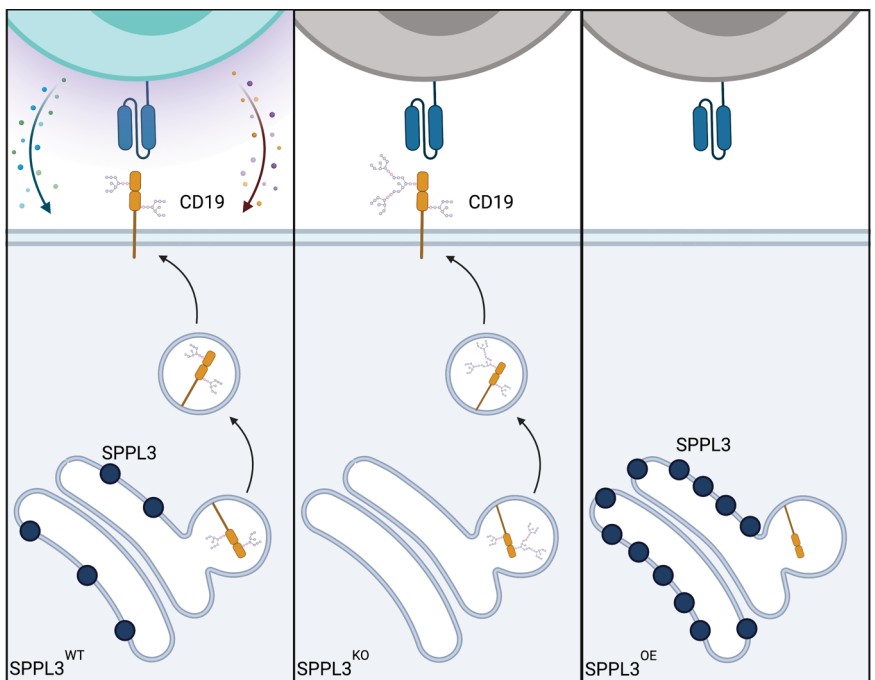

**Fig. 6 Proposed model for the mechanism of glycosylation-mediated antigen escape.** (left) Normal SPPL3 function results in standard glycosylation of CD19. (Middle) Loss of SPPL3 results in increased and altered glycosylation, disrupting the CAR binding epitope. (Right) Increased SPPL3 results initially in decreased glycosylation followed by degradation of CD19, preventing surface presenation.

sparing patients with tumors that are unlikely to respond to ineffective therapies. Gene and transcript-level alterations have previously been shown to impair CAR recognition of CD19 leading to antigen escape. Here we expand the mechanisms that lead to antigen escape to include post-translation modifications of CD19. We found that modulation of SPPL3, either loss or overexpression, resulted in changes to CD19 glycosylation that were both associated with impairment of anti-CD19 CAR T cell function (Fig. 6).

A characteristic feature of malignant transformation is a global alteration in protein glycosylation[25]. This has been specifically extended to N-linked glycosylation of transmembrane proteins[26].

As a result of this intrinsic alteration of surface protein glycosylation, some malignant cells may be predisposed to evade CAR binding. A recent report demonstrated this phenomenon in pancreatic cancer cells, wherein increased cellular glycosylation occurring through an unclear mechanism was associated with decreased CAR efficacy[27]. Interestingly, they found that disruption of N-acetylglucosaminyltransferase V (GnTV, encoded by *MGAT5*), a Golgi-resident glycosyltransferase that is a primary target of SPPL3[17], improved the efficacy of CAR therapy. These studies directly complement the data presented here. The authors of that study did not identify a mechanism by which increased glycosylation suppressed CAR T cell efficacy, or how suppressing

glycosylation improved efficacy. In our study we demonstrate that this is a direct result of manipulating antigen glycosylation. Together, these studies confirm that glycosylation impacts the efficacy of CAR therapy in both solid and hematologic cancers. Another study demonstrated that loss of SPPL3 impaired the efficacy of TCR-driven recognition of cancer antigens by promoting hyperglycosylation of MHC-adjacent glycosphingolipids, thus sterically hindering MHC/TCR interactions[28]. While this interaction is not relevant to MHC-independent CARs, this observation adds further evidence that cancer cell glycosylation is a fundamental regulator of anti-cancer cellular therapy.

Our studies focus on the manipulation of SPPL3 expression, but these data suggest that resistance is not inherent to SPPL3 itself but to its role in regulating CD19 glycosylation. Loss of SPPL3 does not impact the efficacy of CART22, and its over-expression also enables resistance to CART19. As such, the impact of these findings derives from the observation that CD19-targeted CAR T cells are highly-sensitive to antigen glycosylation. There are likely several other cellular pathways that can regulate antigen glycosylation and impact tumor cell sensitivity to CAR therapy. Given its role in cleaving glycosylation regulators, SPPL3 serves a centralized, but indirect, role in regulating CD19 glycosylation, and thus served as an ideal system to identify this mechanism of resistance. These studies compel correlative studies investigating the relationship between antigen glycosylation and the clinical efficacy of antigen-targeted immunotherapies. While evaluation of post-translational modifications is more complicated than standard protein expression studies, our data, as well as other recently published studies, underscore the importance of these studies using patient-derived tissues. Understanding the burden of altered glycosylation in clinical failure will be essential in directing the development of therapies that overcome this mechanism of antigen escape.

## Methods

All research conducted was compliant with relevant ethical research standards. Animal studies were approved by the Washington University Institutional Animal Care and Use Committee, Assurance #A-3381-01.

**Genome-wide knockout screen.** The Brunello sgRNA knockout plasmid library was designed and produced as previously described[20,29] and Nalm6 cells were engineered as previously described[13]. For screening studies, $2 \times 10^8$ Brunello-edited Nalm6 cells were combined with $5 \times 10^7$ CART19 cells or control T cells (effector:target ratio 1:4) and co-cultured in standard culture media; $5 \times 10^7$ control Nalm6 cells were frozen for genomic DNA analysis. After 24 h, cultures were collected, underwent dead cell removal, and were prepared for genomic DNA extraction.

**Genomic DNA extraction, guide sequencing, and analysis.** Performed as described previously[13]. Briefly, cells from screening cultures were flash-frozen as dry pellets and resuspended in 6 mL NK lysis buffer (50 mM Tris, 50 mM EDTA, 1% SDS, pH 8) with Proteinase K followed by incubation at 55 °C overnight. The next day, RNase A was added to the sample, mixed thoroughly, and incubated at 37 °C for 30 min. Samples were cooled and then 2 mL of pre-chilled 7.5 M ammonium acetate was added to precipitate proteins. Samples were vortexed and then centrifuged at ≥4000 × g for 10 min. The supernatant was then transferred to a new 15 mL tube. About 6 mL 100% isopropanol was added to the tube, inverted 50 times, and centrifuged at ≥4000 × g for 10 min. The supernatant was discarded and 6 mL of fresh 70% ethanol was used to wash DNA, followed by centrifugation again at ≥4000 × g for 1 min. The supernatant was discarded, and samples were left to air dry, followed by resuspension with 500uL TE (65 °C for 1 h and at room temperature overnight). The next day, the gDNA samples were vortexed briefly, and gDNA concentration was measured. All gDNA was divided into 100 µL PCR reactions with 5 µg of DNA per reaction. Amplification was performed using Takara ExTaq DNA Polymerase and the default mix protocol with the following PCR program: (95° 2 min, (98° 10 sec, 60° 30 sec, and 72° 30 sec) × 24, 72° 5 min). PCR products were gel purified using the QiaQuick gel extraction kit (Qiagen). The purified, pooled library was then sequenced on a HiSeq4000 with ~5% PhiX added to the sequencing lane. Data were analyzed using standard genome-wide library analysis pipelines as well as customized R scripts[13,30,31].

**CRISPR/Cas9-guide design, genomic engineering, and indel detection.** SPPL3 sgRNAs were designed using Benchling (http://Benchling.com). Six guide RNAs targeting early exons were screened for knockout efficiency and we selected two guides for experimental studies (targeting exon 4, sgRNA sequence: AGACAGATGCTCCAATTGGA; targeting exon 6, sgRNA sequence: CACCATCCATGAGAAGCCAA). Nalm6 and OCI-Ly10 cells were electroporated using the Lonza 4D-Nucleofector Core/X Unit using the SF Cell Line 4D Nucleofector Kit (Lonza). For Cas9 and sgRNA delivery, the ribonucleoprotein (RNP) complex was first formed by combining 10 µg of Cas9 Protein (Invitrogen) with 5 µg of sgRNA. Cells were spun down at 300×g for 10 min and resuspended at a concentration of $3–5 \times 10^6$ cells/µL in the specified buffer. The RNP complex, 100 µL of resuspended cells, and 4 µL of 100 µM IDT Electroporation Enhancer (IDT) were combined and electroporated. After electroporation, cells were cultured at 37 °C for the duration of the experimental procedures. Genomic DNA from electroporated cells was isolated (Qiagen DNeasy Blood & Tissue Kit) and 200–300 ng were PCR amplified using Accuprime Pfx SuperMix or Q5 Mastermix (New England Biolabs) and 10 µM forward/reverse primers flanking the region of interest. Primers were designed such that the amplicon was at a target size ~1 kb. PCR products were gel purified and sequenced, and trace files were analyzed to determine KO efficiency. $R^2$ values were calculated, reflecting the goodness of fit after non-negative linear modeling by TIDE software[32].

**General cell culture.** Unless otherwise specified, cells were grown and cultured at a concentration of $1 \times 10^6$ cells/mL of standard culture media (RPMI 1640 + 10% FCS, 1% penicillin/streptomycin, 1% HEPES, 1% non-essential amino acids) at 37 °C in 5% ambient $CO_2$. For glycosidase treatments, cultures were supplemented with kifunensine (Santa Cruz sc-201634) at a concentration of 16 ng/mL. Jurkat, Nalm6, and OCI-Ly10 parental cell lines were obtained from ATCC. Primary human T cells were purified from human peripheral blood mononuclear cells, obtained commercially from Miltenyi (Catalog #150-000-452).

**Lentiviral vector production and transduction of human cells.** Replication-defective, third-generation lentiviral vectors were produced using HEK293T cells (ATCC ACS-4500). Approximately $10 \times 10^6$ cells were plated in T175 culture vessels in DMEM + 10% FCS culture media and incubated overnight at 37 °C. Eighteen to twenty-four hours later, cells were transfected using a combination of Lipofectamine 2000 (96 µL, Invitrogen), pMDG.1 (7 µg), pRSV.rev (18 µg), pMDLg/p.RRE (18 µg) packaging plasmids and 15 µg of expression plasmid. Lipofectamine and plasmid DNA was diluted in 4 mL Opti-MEM media prior to transfer into lentiviral production flasks. At both 24 and 48 h following transfection, culture media was isolated and concentrated using high-speed ultra-centrifugation (8500×g overnight). For T cell engineering, CD4 and CD8 T cells were isolated from Miltenyi PBMC packs and combined at a 1:1 ratio, activated using CD3/CD28 stimulatory beads (Thermo Fisher) at a ratio of 3 beads/cell, and incubated at 37 °C overnight. The following day, CAR lentiviral vectors were added to stimulatory cultures at an MOI of 3. Beads were removed on day 6 of stimulation, and cells were counted daily until growth kinetics and cell size demonstrated they had rested from stimulation. For cancer cell engineering, vectors were combined with cells at an MOI of 2.

**Co-culture assays.** For cytotoxicity assays, CAR T cells were combined with target cells at various E:T ratios and co-cultures were evaluated for an absolute count of target cells by flow cytometry. All co-cultures were established in technical triplicate. Cultures were maintained at a concentration of 1e6 total cells/mL. For re-exposure assays, CAR + T cells were sorted by fluorescence-assisted cell sorting using a truncated CD34 selection marker encoded in the CAR plasmid backbone. T cells were then recombined with target leukemia cells at an effector:target ratio of 1:4 and killing was measured as described. For activation marker studies, CAR or Jurkat T cells and Nalm6 cells were combined at an E:T ratio of 1:4 and evaluated by flow cytometry the following day. Jurkat cells were engineered to express a dual fluorescence reporter system indicating activation of transcription factor activity as previously described[33]. For degranulation assays (CD107a assessment), T cells were combined with Nalm6 as described and combined with an antibody cocktail of CD107a-PECy7 (clone H4A3, Biolegend 328607, diluted 1:100) and stimulatory antibodies against CD28 (eBiosciences 16-0288-81, diluted 1:50) for 1 h. Intracellular protein transport was halted by the addition of GolgiStop (BD Biosciences 554724) and cells were incubated for an additional 3 h. Cells were then harvested and stained for CD34 (BD 555824, diluted 1:50) and analyzed by flow cytometry.

**Xenograft mouse models.** About 6–10-week-old female NOD-SCID-γc$^{-/-}$ (NSG) mice were obtained from the Jackson Laboratory and maintained in pathogen-free conditions with standard dark/light cycles and ambient temperature and humidity. Each experimental group contained between 4–7 mice. Animals were injected via tail vein with $1 \times 10^6$ WT or SPPL3$^{KO}$ Nalm6 cells in 0.2 mL sterile PBS. On day 6 after tumor delivery, $1 \times 10^6$ T cells were injected via tail vein in 0.2 mL sterile PBS. Animals were monitored for signs of disease progression and overt toxicity, such as xenogeneic graft-versus-host disease, as evidenced by >10% loss in body weight, loss of fur, diarrhea, conjunctivitis and disease-related hind limb paralysis. Disease burdens were monitored over time using the Spectral

Instruments AMI bioluminescent imaging system. Animal were sacrificed when tumor burden reached >1 × 10^10 photons/sec/steradian/cm^2. No animal exceeded this burden. Animal studies were approved by the Washington University Institutional Animal Care and Use Committee, Assurance #A-3381-01.

**Flow cytometry**. Cells were resuspended in FACS staining buffer (PBS + 3% fetal bovine serum) using the following antibodies: CD3 (clone OKT3, BD Biosciences 555342, diluted 1:50), PD1 (clone EH12.2H7, BioLegend, 329928, diluted 1:80), Tim3 (clone 7D3, BD Biosciences 565566, diluted 1:100), CD22 (clone HIB22, BD Biosciences 562860, diluted 1:100), CD19 (clone FMC63, Novus Biologicals 52716, diluted 1:80-1:32000; clone HIB19, BD Biosciences 555413, diluted 1:50), CD107a-PECy7 (clone H4A3, Biolegend 328607, diluted 1:100), CD34 APC (BD 555824, diluted 1:50). CARs transduction was evaluated by staining for a truncated CD34 selection marker located downstream of a P2A ribosomal skip sequence from the CAR transgene. Data were acquired on an Attune NxT cytometer (Thermo). All data analysis was performed using FlowJo 9.0 software (FlowJo, LLC). The gating strategy can be found in Supplementary Fig. 5.

**Western blotting**. Nalm6 and OCI-Ly10 cells were counted and 5 × 10^6 cells were washed in cold PBS. Cell pellets were resuspended in RIPA lysis buffer supplemented with phosphatase and protease inhibitors and incubated on ice for 15 min, followed by centrifugation at 14,000 × g for 15 min. Lysate concentration was quantified using the Pierce BSA Protein Assay Kit (Thermo), combined with 4x LDS buffer, denatured at 100 °C for 10 min, and then reduced to a final concentration of 20% beta-mercaptoethanol. 10–20 ug of protein was loaded into each well of a Bis-Tris gel (either 4–12% gradient, 6%, or 15%) and proteins were separated using standard electrophoresis followed by transfer to nitrocellulose membranes. Proteins were labeled with SPPL3 (EMD Millipore #MABS1910), CD19 (#3574), CD22 (#67434), Actin (#4970) or GAPDH (#2118, all from Cell Signaling, all diluted 1:1000), followed by secondary antibody staining using either anti-rabbit (Cell Signaling #7074, diluted 1:5000), or anti-mouse (Cell Signaling #7076, diluted 1:5000) HRP-linked antibodies followed by visualization. Cell components were isolated using the Cell Signaling Cell Fractionation Kit (#9308). Unprocessed plots can be found in Source Data Files.

**Immunoprecipitation**. WT and SPPL3^KO Nalm6 cells were harvested in RIPA lysis buffer supplemented with protease inhibitors and incubated on ice for 10 min, followed by centrifugation to clear debris (14,000 × g, 10 min, 4 °C). Lysate concentration was quantified using Bradford reagent and normalized to 1–3 mg total protein per 1 mL. Normalized lysates were cleared by incubation with protein A beads (Invitrogen) on a rotator (1 h, 4 °C) and then incubated with 2 uL anti-CD19 antibody (clone FMC63, Novus Biologicals) on a rotator (overnight, 4 °C). Antibody-conjugated lysates were incubated with protein A beads on a rotator (4 h, 4 °C). Beads were then washed three times in RIPA buffer and reserved for lectin blot analysis.

**Lectin blotting**. Immunoprecipitated proteins were removed from the beads using 2x Sample Buffer (20% (v/v) glycerin, 8% (w/v) SDS, 7,5 % (w/v) DTT, 0,5 M Tris pH 6,8 supplemented with bromophenol blue) and were heated at 65 °C for 10 min. Samples were then loaded on 8% SDS gels and the proteins were separated using standard electrophoresis followed by transfer to PVDF membranes. The membranes were first incubated in 5% (w/v) BSA in PBS-T overnight. Biotin and streptavidin blocking was performed using a commercial kit (Vector Laboratories, California, USA) according to the manufacturer's instructions. Membranes were incubated with the respective biotinylated lectin (0.5 µg/ml in lectin buffer (10 mM HEPES, pH 7.5, 150 mM NaCl, 0.1 mM CaCl₂, 0.08% (w/v) NaN₃, and only for ConA 10 µM MnCl2) Vector Laboratories, California, US) for 1 h at room temperature. Following three washing steps (10 min each with PBS-T), blots were incubated with a streptavidin-HRP conjugate (0.5 µg/ml, diluted in PBS-T) at room temperature for 30 min. After three additional washing steps, blots were developed using conventional ECL chemistry (GE Healthcare, Chalfont St Giles, UK).

**Protein modeling**. Prediction of CD19 structure was performed using Phyre2 web portal for protein modeling, prediction, and analysis[21] and analyzed and visualized using UCSF Chimera[34]. Sequences used for predictive modeling and analysis were derived from the Protein Database entry 6AL5.

**Statistical analysis**. Statistical analysis performed using GraphPad Prism v9 unless otherwise indicated. All data presented were representative of independent experiments using T cells derived from between two to five independent donors, except for the CRISPR knockout screen (performed once with four biological replicates). All cytotoxicity studies and flow-based protein expression studies were performed in technical triplicate. Comparisons between two groups were performed using either a two-tailed unpaired Student's t-test. Comparisons between more than two groups were performed by two-way analysis of variance (ANOVA) with Bonferroni correction for multiple comparisons. All results are represented as mean ± standard error of the mean (s.e.m.).

**Reporting summary**. Further information on research design is available in the Nature Research Reporting Summary linked to this article.

## Data availability
Guide library sequencing data were available from Gene Expression Omnibus (GEO) using the accession number GSE130663. Sequences used for predictive modeling and analysis were derived from the Protein Database entry 6AL5. The remaining data are available with the Article, Supplementary Information, or Source Data File. Source data are provided with this paper.

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

## Acknowledgements

The authors wish to thank Ophir Shalem (The Children's Hospital of Philadelphia) and Jonathan Schug (University of Pennsylvania) for substantial feedback in the development and execution of the genome-wide knockout screen and sequencing, Julie Ritchey, Alun Carter, and Matthew Cooper (Washington University School of Medicine) for intellectual and technical support, Peter Steinberger (Medical University of Vienna) for the Jurkat triple parameter reporter cell line and Elena Orlando (Novartis) for useful discussions. This work was supported by NIH K08CA237740 and The Amy Strelzer Manasevit Research Grant from the Be The Match Foundation (N.S.), NIH K08CA212299 (A.M.G.) as well as by the German Research Foundation (DFG, Deutsche Forschungsgemeinschaft) grants 263531414/FOR 2290 and 254872893/FL 635/2-3 (R.F.) Illustrations made using BioRender.com.

## Author contributions

A.H., J.H.L., R.F., and N.S. designed and oversaw the research and wrote the manuscript. A.H., J.H.L., A.R.H., A.P., M.H.-K., M.E.S., Y.-S.H., J.L., J.C., H.H., J.M.W., A.M.G., and N.S. performed the research. B.D., M.R., and S.G. provided significant technical and conceptual contributions. K.E.H. and M.D.W. performed bioinformatical analyses. All authors reviewed the manuscript.

## Competing interests

N.S. holds patent 15/567,156 (Methods for Improving the Efficacy and Expansion of Chimeric Antigen Receptor-Expressing Cells); N.S., M.R., and S.G. hold patent 17/058,163 (Combination Therapy with Chimeric Antigen Receptor Therapies); M.R. and S.G. hold patent 16/256,731(CD20 therapies, CD22 therapies, and combination therapies with CD19 chimeric antigen receptor-expressing cells) which are related to investigational and commercial CAR T cell products. The remaining authors declare no competing interests.
