## [Peer Review File · Nature Communications]

Reviewers' Comments:

Reviewer #2:

Remarks to the Author:

In the article by Heard et al the authors investigate underlying mechanisms to explain resistance against CD19 CAR T therapy. The authors make a highly compelling and convincing case that SPPL3 is involved through the use of KO and overexpression studies. Throughout the article (and title) the authors claim that altered glycosylation of CD19 is the critical regulator, but by no means do they provide true evidence for this. Yes, SPPL3 has many glycosyltransferase targets, so altered glycosylation seems likely, but also alternative targets have been identified for SPPL3 (Kuhn Mol Cell Prot 2015). In my opinion this is the crucial point of this article.

How sure are the authors that CD19 is indeed differentially N-glycosylated (I only see one western blot, no actual glycan profiling what so ever)? Do they really prove that this altered glycosylation is causal and not a bystander effect? The authors even refer to a paper by Klesmith et al that FMC63 binding is not glycan-dependent. This of course contradicts all their claims regarding the antigen glycosylation. Moreover, the Tyr260 mutant the authors mention could even hint to a role for O-glycosylation.

Thus, to uphold that CD19 glycosylation is actually the causal factor determining the CAR T resistance the authors should:

- Perform more extensive glycan profiling of the CD19 in their WT and KO cell lines, ideally via LC/MS-MS to identify the altered glycans on CD19 (and to prove that glycosylation is indeed altered and not that there is just more coverage of CD19 with the same glycans). Alternatively, lectins blot, could provide indications on the altered glycan epitopes on immunoprecipitated CD19.
- Next, the authors should KO/knock down relevant glycosyltransferases to revert the SPPL3 phenotype. Some prime candidates have been already identified, which could form a starting points for these assays, like MGAT5/GnTV or ST6Gal1. Recently, Jongsma et al published (Immunity 2021) that SPPL3 regulates glycosphingolipid synthesis, which in tumor cells shield MHC molecules, thus preventing proper anti-tumor immunity. Have the authors considered this mechanism? Does KO of B3GNT5 revert their phenotype as well?
- If SPPL3 and altered glycosylation are involved, the use of glycosylation inhibitors (sialic acid analogues, kifunensine or maybe even O-benzylGalNAc) should be able to induce the resistance phenotype as well.

Reviewer #3:

Remarks to the Author:

Chimeric antigen receptor (CAR) T cell therapy has been demonstrated to be highly effective in various cancer treatments, especially in treating B cell malignancies. However, one of the major obstacles to successful CAR T therapy is relapse. About one-third of patients have tumor relapse, most of which occurs within one year of CAR-T treatment. Therefore, dissecting the molecular mechanism by which the tumor cells escape the immunosurveillance mediated by transferred CAR-T cells is clinically important. In the manuscript entitled "Antigen glycosylation is a critical regulator of CAR T cell efficacy", Amanda et al. performed a CRISPR-based genome-wide genetic screen to identify the potential genes whose deletion in target cells could lead to CAR-T killing resistance. The authors found that SPPL3 was a top hit (Rank #2). Mechanistically, SPPL3 mediates modification of CD19 glycosylation and either KO or overexpression could result in the poor recognition of FMC63, an antibody widely used in the current CD19-CART product. The study tells a clean story, which reveals a possible mechanism of antigen escape for CAR-T therapy. The manuscript is also well written. However, the current manuscript has the following concerns:

1. The authors identified a possible way that tumor cells could be resistant to CAR-T killing. However, there is no evidence showing that if the same observation occurs in the body of the treated patients.
2. The dysfunction of CAR-T cells induced by residential antigen described in Figure 2i is very interesting. It would be more informative if authors could test if CAR-T failure depends on CD28 or 41BB costimulatory domain within CAR constructs.

Minor concerns: I didn't see the ref 29 was cited in the main text.

Reviewer #4:

Remarks to the Author:

Singh and colleagues report on the ability of under- or over-expression of SPPL3 to modulate glycosylation of CD19 thereby hindering CD19-directed chimeric antigen receptor (CAR) T cell therapy in cell culture. This study represents a unique mode of antigen escape in CAR T cell therapy and thus is an important advance in fundamental understanding with potential clinical implications.

A genome-wide screen identifies an SPPL3 knockout as strongly hindering of CAR T cell-mediated death of Nalm6 cells. Clonal knockout analysis reveals diminished CAR T cell-mediated killing of SPPL3 knockouts in two cell lines. Reduced T cell activation and T cell dysfunction are demonstrated in response to SPPL3 knockout. CD19 hyperglycosylation is demonstrated via SDS-PAGE and glycosylase assay and shown to hinder CD19/antibody binding. Lack of impact on CD22 glycosylation and CD22 CAR T cell activity support a glycosylation-driven mechanism of action. SPPL3 overexpression yields hypoglycosylation and impaired CAR activity, which is consistent with the mechanism of reduced CAR T cell engagement upon improper CD19 post-translational modification.

The thorough suite of experiments are designed well. The results are consistent and support the stated conclusions. The writing is clear. Thus, the manuscript provides a valuable contribution to the field.

One simple edit is encouraged: It is not clear if the CRISPR screen (Figure 1b) was performed as a new experiment or if previous data ("as we reported previously... [ref 13]") are plotted in a new format. Either case is acceptable, but it should simply be more clear; it's not currently evident if the "as we reported previously... [ref 13]" refers to obtaining a matching result (FADD as the most enriched gene) or restating the previous data.

Reviewer #5:

Remarks to the Author:

Loss of CD19 expression (either epitope or entire protein loss) in B cell malignancies is a clinically relevant mechanism causing disease recurrence after CD19.CAR-T therapy. It is definitely possible that tumor cells, despite expressing the targeted antigen, can evolve and develop non checkpoint-inhibitor mediated intrinsic mechanisms of resistance to the cytotoxic effects of CAR-T cells. However, the clinical impact of these intrinsic mechanisms of resistance remains unknown or limited to anecdotal observations, such as the previously FADD-dependent mechanism, that require validation in large cohort of patients. Using a previously described Brunello genome-wide RNA library, Hear and colleagues report that either loss or overexpression of SPPL3, an enzyme implicated in protein glycosylation, in tumor cells causes alteration of CD19 glycosylation and impairs the recognition by CAR-T cells.

Major comments

- 1) Lack of in vivo data in mouse models supporting lack of activity of CAR-T cells when SPPL3 is altered. Data in vitro presented in Figure 1 for the SPPL3 KO suggest that this defect may be corrected by the increasing the number of CAR-T cells used and may not be relevant in vivo using therapeutic doses of CAR-T cells in mice.
- 2) Lack of clinical evidences that SPPL3 is involved in causing leukemia/lymphoma relapse in patients relapsing with CD19+ tumors.

Response to Reviewers

Reviewer #2

Overall:

In the article by Heard et al the authors investigate underlying mechanisms to explain resistance against CD19 CAR T therapy. The authors make a highly compelling and convincing case that SPPL3 is involved through the use of KO and overexpression studies. Throughout the article (and title) the authors claim that altered glycosylation of CD19 is the critical regulator, but by no means do they provide true evidence for this. Yes, SPPL3 has many glycosyltransferase targets, so altered glycosylation seems likely, but also alternative targets have been identified for SPPL3 (Kuhn Mol Cell Prot 2015). In my opinion this is the crucial point of this article.

Major comment 1:

How sure are the authors that CD19 is indeed differentially N-glycosylated (I only see one western blot, no actual glycan profiling what so ever)? Do they really prove that this altered glycosylation is causal and not a bystander effect?

Response:

We appreciate the Reviewer's point that we had not definitively proven that glycosylation is the causative mechanism of resistance. In the responses to the specific experimental recommendations we go into further detail about our work to address this more directly in response to these helpful suggestions.

Our original conclusion that altered CD19 glycosylation was responsible for the observed resistance to CD19 CAR T cell cytotoxicity relied on the data from our CD22 studies and glycosylation inhibitor studies. CD22 did not demonstrate any evidence of altered post-translational modification, as assessed by immunoblot, in the context of SPPL3 loss. Disruption of SPPL3 also did not impair binding of anti-CD22 antibodies or engender a resistance phenotype to CD22 CAR T cell therapy. If resistance to CD19 CAR T cells was the result of a distinct, CD19-independent effect of SPPL3 loss (a bystander effect the Reviewer refers to), we speculated that cells should also be resistant to CD22 CAR T cells via the same mechanism.

Detailed below are our additional interrogations including glycan profiling, treatment of cells with the glycosylation inhibitor kifunensine and mutation of individual asparagine residues to render them glycosylation insensitive that we posit definitively confirm that CD19 glycosylation is the causative mechanism of CAR T cell failure.

Major comment 2:

The authors even refer to a paper by Klesmith et al that FMC63 binding is not glycan-dependent. This of course contradicts all their claims regarding the antigen glycosylation.

Response:

In Klesmith et al. (*Biochemistry* 2019) the authors demonstrate that inhibition of N-glycosylation does not impact FMC63 binding. We respectfully clarify that the findings in Klesmith et al. do not contradict our claims. This paper does not address the impact of hyperglycosylation of CD19 on FMC63 binding, which constitutes the majority of our findings.

In the first submission, we did demonstrate that over-expression of SPPL3 resulted in hypoglycosylation of CD19, loss of surface antibody binding and resistance to CD19 CAR T cell therapy. We originally speculated that this may be a result of hypoglycosylation itself or intracellular retention of CD19 (reflected in our Schematic in Figure 5), which was more likely in light of data presented in Klesmith et al. We have since performed additional experiments using cells engineered to over-express SPPL3. Our original data demonstrated that transgenic expression initially results in hypoglycosylation, as reflected in

the western blot from lysates generated 5 days after engineering. Given the kinetics of failed CD19 antibody binding (progressing over time, **Figure 5a**), we speculated that hypoglycosylation that results from SPPL3 over-expression caused intracellular retention of CD19 (as has been shown for other structural disruptions in Bagashev A. et al. *Mol Cell Biol*, 2018). To evaluate this hypothesis, we performed fractionated cellular lysis and immunoblot for CD19. To our surprise, we found that maintained over-expression of SPPL3 does not cause intracellular retention, but caused complete loss of CD19 protein (**Figure 5b**). We confirmed that SPPL3 over-expressing cells continue to transcribe *CD19* by quantitative RT-PCR at the same level as wild-type or *SPPL3*^{KO} cells (**Supplementary Figure 4c**). Previous work has demonstrated that hypoglycosylation impairs CD19 surface expression (Mortales C. et al. *J Immunol*, 2020), consistent with our findings. Based on these observations along with the contextual literature, we predict that over-expression of SPPL3 results in alterations that destabilize protein structure and result in degradation.

In summary, our additional studies demonstrate that over-expression of SPPL3 results in resistance to CAR therapy by initially causing hypoglycosylation and subsequently leading to protein loss. These findings are presented in **Figure 5** and **Supplementary Figure 4**.

Major comment 3:

Moreover, the Tyr260 mutant the authors mention could even hint to a role for O-glycosylation.

Response:

To avoid extensive discussion of data that had not been finalized in the peer review process, we tried to keep discussion of the Tyrosine 260 deletion minimal. We realize now that this led to miscommunication, and so have removed reference to this distinct manuscript this from our Discussion to allow those data to stand independently. To the Reviewer's comment, CD19 is not believed to be O-glycosylated but only N-glycosylated (Klesmith et al. *Biochemistry* 2019 and <https://www.uniprot.org/uniprot/P15391>), which is why we have focused our attention on this mechanism. Confidentially, further studies related to deletion Y260 reveal that loss of this residue prevents surface expression as the mechanism of resistance to CAR therapy.

Major Comment 4:

Thus, to uphold that CD19 glycosylation is actually the causal factor determining the CAR T resistance the authors should:

Perform more extensive glycan profiling of the CD19 in their WT and KO cell lines, ideally via LC/MS-MS to identify the altered glycans on CD19 (and to prove that glycosylation is indeed altered and not that there is just more coverage of CD19 with the same glycans). Alternatively, lectins blot, could provide indications on the altered glycan epitopes on immunoprecipitated CD19.

Response:

As the Reviewer suggests, LC/MS-MS to interrogate glycan identity is a highly-specialized technique with which few investigators have experience. We instead elected to pursue lectin blots of immunoprecipitated CD19 protein from WT cells or *SPPL3*^{KO} cells. We used concanavalin A (ConA) binding to identify the presence of mannose glycans and phytohaemagglutinin-L (PHA-L) binding to detect branched glycans. As shown in **Supplementary Figure 2c**, loss of SPPL3 resulted in similar ConA binding, suggesting no change in core glycans. Concurrently we observed a reduction in PHA-L binding. This could occur via either (1) a reduction in branched glycans or (2) masking of the PHA-L binding epitope due to over-branching. In light of our data demonstrating reduction in size of CD19 only with

PNGaseF and not EndoH (**Figure 3b**), which confirms presence of branched glycans, these data confirm that loss of SPPL3 significantly alters CD19 branched glycan structure.

Major Comment 5:

Next, the authors should KO/knock down relevant glycosyltransferases to revert the SPPL3 phenotype. Some prime candidates have been already identified, which could form a starting points for these assays, like MGAT5/GnTV or ST6Gal1. Recently, Jongsma et al published (Immunity 2021) that SPPL3 regulates glycosphingolipid synthesis, which in tumor cells shield MHC molecules, thus preventing proper anti-tumor immunity. Have the authors considered this mechanism? Does KO of B3GNT5 revert their phenotype as well?

Response:

We appreciate that assessment of an SPPL3 target, such as MGAT5, would be a more direct experiment to confirm a glycosylation-dependent mechanism of resistance. We attempted to knockout *MGAT5* in our *SPPL3*^{KO} Nalm6 cells by CRISPR/Cas9-mediated genomic disruption using four independent guide RNAs, but this resulted in overwhelming and unexpected cell death with outgrowth of only *MGAT5*-intact cells. We then attempted to knockdown transcribed *MGAT5* using three independent siRNAs, but these also did not yield sufficient outgrowth of *MGAT5*-depleted cells. After discussions with our collaborator Dr. Regina Fluhrer, a glycobiologist, we speculate that *MGAT5* is an essential protein for normal cellular function of our Nalm6 cells, as her lab has seen these survival disadvantage with *MGAT5* disruption in other cell lines.

Since initial submission of our manuscript, a report was published demonstrating that inhibition of *MGAT5* in pancreatic cancer cells enhances the efficacy of CAR T cells (Greco et al. *Science Translational Medicine*, 2022). The conclusion of this study is that global hyperglycosylation (occurring via unclear mechanisms) impairs CAR T cell efficacy, and reducing glycosylation is of therapeutic benefit. These data reflect the exact experiments suggested by the reviewer, and corroborate the data presented in our manuscript in an orthogonal and complementary manner. Notably, the authors in Greco et al. admit that they did not identify the mechanism by which hyperglycosylation results in resistance to CAR therapy. In this updated manuscript, we present compelling data that increased glycosylation of the target antigen itself is the mechanistic etiology.

While inhibition of *MGAT5* in *SPPL3*^{KO} cells would be a form of a “rescue” experiment, it would not specifically isolate the role of altered CD19 glycosylation in resistance, but instead interrogate how rescuing the global (*MGAT5*-dependent) glycoproteome impacts resistance. To address this directly, we performed mutational studies on the asparagine residues in closest proximity to the anti-CD19 CAR binding site to confirm that altered CD19 glycosylation, and not another mechanism, is responsible for resistance (see Response to Comment #6 below). In light of these studies, as well as the fact that CAR activity is MHC-independent (in contrast to the model used in Jongsma et al.), we did not pursue studies interrogating the role of B3GNT5.

Major Comment 6:

If SPPL3 and altered glycosylation are involved, the use of glycosylation inhibitors (sialic acid analogues, kifunensine or maybe even O-benzylGalNAc) should be able to induce the resistance phenotype as well.

Response:

The Reviewer suggests an important experiment. We previously tried to inhibit CD19 glycosylation using tunicamycin and benzyl-2-acetamido-2-deoxy- α -D-galactopyranoside to inhibit N and O glycosylation, respectively. These inhibitors were highly toxic to our cell lines at the concentrations required for

effective inhibition and thus prohibited any evaluation of sensitivity to CAR T cell killing. At the Reviewer's recommendation, we cultured *SPPL3*^{KO} cells in varying concentrations of kifunensine (0-16ng/mL) for 10 days and saw no toxicity. Western blot analysis revealed a reduction in CD19 molecular weight (**Figure 3d**), confirming that kifunensine treatment reduced CD19 glycosylation. Flow cytometric analysis of these cells revealed enhanced binding of FMC63 in the setting of deglycosylation (**Supplementary Figure 3d**). Consistent with this enhanced binding, co-culture of kifunensine treated *SPPL3*^{KO} cells with CD19 CAR T cells rescued the resistance phenotype, rendering them as sensitive to CAR T cell killing as wild-type cells (**Figure 3e**). Together, these data support our conclusion that cellular hyperglycosylation is the causative etiology of resistance to CAR T cell cytotoxicity.

As kifunensine treatment does not only inhibit glycosylation of CD19, but also other surface glycoproteins, it was feasible that reduced glycosylation of a distinct protein (or proteins) was responsible for the rescued sensitivity to CAR T cell killing. While our studies with CD22 would strongly argue against this (again, if resistance was not directly related to CD19 glycosylation, *SPPL3*^{KO} cells should have also been resistant to CART22), we undertook a series of studies to specifically evaluate the contribution of the asparagine residues in close proximity to the FMC63 binding site. As demonstrated in **Figure 4a**, evaluation of the previously solved CD19 structure reveals two asparagine residues (114 and 125) in close proximity to the FMC63 binding site. We generated an *SPPL3*^{KO} cell line that was also deficient for CD19 (Dual KO). We then introduced either WT CD19 or CD19 mutants in which either N114 or N125 had been converted to glutamic acid residues (N114Q and N125Q), preventing their glycosylation. These transgenic CD19 molecules were encoded on lentiviral expression plasmids that also contained a CD34 selection marker. Using flow cytometry-assisted sorting, we purified cells to ensure 100% of cells expressed CD34 and that the level of CD34 expression was equivalent. This generated cell populations that have the same quantity of CD19 on the surface and only express the CD19 molecules we introduced. We then evaluated binding of FMC63 and sensitivity to CART19 killing. As demonstrated in **Figure 4b**, FMC63 binding was equivalent between WT Nalm6 cells and *SPPL3*^{KO}*CD19*^{KO} Nalm6 cells engineered to express WT CD19. We found that mutation at N114 did not alter FMC63 binding. Mutation of N125, however, enhanced FMC63 binding to the level seen in wild-type (*SPPL3* and CD19 intact) Nalm6 cells. Consistent with the enhanced FMC63 binding, we found that *SPPL3*^{KO} cells engineered to express CD19*N125Q were now sensitive to CART19 killing. Notably, N114 is not normally glycosylated while N125 is; increased glycan quantity at N125, as opposed to *de novo* glycosylation of N114 is consistent with known *SPPL3* function.

These studies strongly implicate altered glycosylation of CD19, specifically at residue N125, as the causative mechanism of resistance to CART19.

Reviewer #3

General

Chimeric antigen receptor (CAR) T cell therapy has been demonstrated to be highly effective in various cancer treatments, especially in treating B cell malignancies. However, one of the major obstacles to successful CAR T therapy is relapse. About one-third of patients have tumor relapse, most of which occurs within one year of CAR-T treatment. Therefore, dissecting the molecular mechanism by which the tumor cells escape the immunosurveillance mediated by transferred CAR-T cells is clinically important. In the manuscript entitled "Antigen glycosylation is a critical regulator of CAR T cell efficacy", Amanda et al. performed a CRISPR-based genome-wide genetic screen to identify the potential genes whose deletion in target cells could lead to CAR-T killing resistance. The authors found that SPPL3 was a top hit (Rank #2). Mechanistically, SPPL3 mediates modification of CD19 glycosylation and either KO or overexpression could result in the poor recognition of FMC63, an antibody widely used in the current CD19-CART product. The study tells a clean story, which reveals a possible

mechanism of antigen escape for CAR-T therapy. The manuscript is also well written. However, the current manuscript has the following concerns:

Major Comment 1:

The authors identified a possible way that tumor cells could be resistant to CAR-T killing. However, there is no evidence showing that if the same observation occurs in the body of the treated patients.

Response:

The Reviewer brings up an essential point. The clinical relevance of altered glycosylation is highly-relevant but extremely difficult to evaluate. To successfully determine if altered CD19 glycosylation causes therapeutic failure in patients, we need to directly compare CD19 glycosylation in pre-treatment to relapsed or refractory samples, or correlate pre-treatment CD19 glycosylation status with clinical outcomes. In discussion with our clinical colleagues at Wash U, as well as Penn and MD Anderson, it has become clear that the necessary tissues do not currently exist in quantities needed to perform these studies.

Pre-treatment biopsies are often collected months to years prior to treatment with CAR T cells, generating significant variability in the cells that were collected and the cells that are treated. Tumor heterogeneity, both spatial and temporal, introduces further variability. Acquisition of relapsed samples is further complicated for patients with lymphoma, as these tissues are rarely collected after therapeutic failure of CAR T cells and collection requires an interventional procedure. If we were able to collect a repository of these samples prospectively, they would need to undergo either immunoblotting to examine changes in CD19 molecular weight and lectin blotting, which would require a substantial number of purified CAR+ cells, or glycan mass spectrometry, a complex technique with which few investigators have expertise.

We have spoken to our colleagues at the University of Pennsylvania and MD Anderson Cancer Center, two of the primary sites for the large CAR T cell trials that led to FDA approvals of commercial products. Neither center has performed proteomic studies, nor do they have the tissue available to perform these studies in a robust way at this time. We are building the tools to perform proteomic interrogation of clinical samples in collaboration with Dr. Benjamin Garcia (using nanoscale LC-MS), a leader in post-translation modification proteomics, but these assays will take substantial time to optimize.

As a surrogate measure of altered glycosylation machinery, we collaborated with our colleagues Drs. Marco Ruella and Stephan Schuster from the University of Pennsylvania and Dr. Elena Orlando at Novartis to interrogate bulk RNA sequencing data on pre-treatment biopsies from patients enrolled on an institutional trial of CD19-directed CAR T cells for patients with lymphoma. We have included these data as **Reviewer Figure 1**.

[Redacted]

We find these data very interesting and, in context of the data presented in

the manuscript certainly believe they compel further study, but for these reasons we are hesitant to make any conclusions from these studies and have not included these data in the manuscript.

We have also addressed the relevance of this question directly in the Discussion.

Major Comment 2:

The dysfunction of CAR-T cells induced by residential antigen described in Figure 2i is very interesting. It would be more informative if authors could test if CAR-T failure depends on CD28 or 41BB costimulatory domain within CAR constructs.

Response:

We performed a similar experiment to one previously presented using CD28-based CD19 CAR T cells. We again observed the development of T cell functional impairment after prolonged exposure to SPPL3-deficient Nalm6 cells, with an inability to kill wild-type targets on re-exposure. These findings indicate that this is not a co-stimulatory domain-dependent phenomenon. We have added these data to **Supplementary Figure 2a**.

Minor Comment 1:

I didn't see the ref 29 was cited in the main text.

Response:

We regret this error. The Discussion has been modified to include the relevance of this manuscript to our findings.

Reviewer #4

General:

Singh and colleagues report on the ability of under- or over-expression of SPPL3 to modulate glycosylation of CD19 thereby hindering CD19-directed chimeric antigen receptor (CAR) T cell therapy in cell culture. This study represents a unique mode of antigen escape in CAR T cell therapy and thus is an important advance in fundamental understanding with potential clinical implications.

A genome-wide screen identifies an SPPL3 knockout as strongly hindering of CAR T cell-mediated death of Nalm6 cells. Clonal knockout analysis reveals diminished CAR T cell-mediated killing of SPPL3 knockouts in two cell lines. Reduced T cell activation and T cell dysfunction are demonstrated in response to SPPL3 knockout. CD19 hyperglycosylation is demonstrated via SDS-PAGE and glycosylase assay and shown to hinder CD19/antibody binding. Lack of impact on CD22 glycosylation and CD22 CAR T cell activity support a glycosylation-driven mechanism of action. SPPL3 overexpression yields hypoglycosylation and impaired CAR activity, which is consistent with the mechanism of reduced CAR T cell engagement upon improper CD19 post-translational modification.

The thorough suite of experiments are designed well. The results are consistent and support the stated conclusions. The writing is clear. Thus, the manuscript provides a valuable contribution to the field.

Major Comment 1:

One simple edit is encouraged: It is not clear if the CRISPR screen (Figure 1b) was performed as a new experiment or if previous data ("as we reported previously... [ref 13]") are plotted in a new format. Either case is acceptable, but it should simply be more clear; it's not currently evident if the "as we reported previously... [ref 13]" refers to obtaining a matching result (FADD as the most enriched gene) or restating the previous data.

Response:

We regret this was not clear. These data are indeed from the previous screen that was published in reference 13. We have plotted them in a new format here to highlight SPPL3. We have modified the Results to reflect this.

Reviewer #5

General:

Loss of CD19 expression (either epitope or entire protein loss) in B cell malignancies is a clinically relevant mechanism causing disease recurrence after CD19-CAR-T therapy. It is definitely possible that tumor cells, despite expressing the targeted antigen, can evolve and develop non checkpoint-inhibitor mediated intrinsic mechanisms of resistance to the cytotoxic effects of CAR-T cells. However, the clinical impact of these intrinsic mechanisms of resistance remains unknown or limited to anecdotal observations, such as the previously FADD-dependent mechanism, that require validation in large cohort of patients. Using a previously described Brunello genome-wide RNA library, Hear and colleagues report that either loss or overexpression of SPPL3, an enzyme implicated in protein glycosylation, in tumor cells causes alteration of CD19 glycosylation and impairs the recognition by CAR-T cells.

Major Comment 1:

Lack of in vivo data in mouse models supporting lack of activity of CAR-T cells when SPPL3 is altered. Data in vitro presented in Figure 1 for the SPPL3 KO suggest that this defect may be corrected by the increasing the number of CAR-T cells used and may not be relevant in vivo using therapeutic doses of CAR-T cells in mice.

Response:

We concur, this is an important additional indicator of resistance. Using our established immunodeficient NSG model of systemic ALL, we engrafted mice with either WT or *SPPL3*^{KO} Nalm6 cells and treated with either nothing (n=4 in each group) or CD19-targeted CAR T cells (n=7 in each group). As reflected in **Figure 1e** and **Supplementary Figure 1e**, mice with SPPL3-deficient tumors that received CAR T cells had similar tumor growth to those mice that received no treatment, while CAR T cells mediated a significant anti-tumor effect in mice with WT tumors. This resulted in a significant improvement in animal survival (**Figure 1f**). The reviewer brings up an important point that higher doses of T cells can often mask biologically relevant differences in sensitivity to CAR T cell efficacy; we noticed a similar phenomenon previously (Singh et al. *Cancer Discovery*, 2020). Indeed, high enough doses of CAR T cells can even eradicate antigen-low tumors that have escaped from clinical doses of CAR T cells. We are certainly mindful of the limitations of our models and aim to corroborate our *in vitro* and *in vivo* findings with the clinical correlative studies described in response to Reviewer 3 and below.

Major Comment 2:

Lack of clinical evidences that SPPL3 is involved in causing leukemia/lymphoma relapse in patients relapsing with CD19+ tumors.

Response:

We do greatly appreciate the importance of clinical relevance. As discussed in the Response to Reviewer 3 Comment 1, we posit that the most relevant question is direct assessment of CD19 glycosylation, which could be altered by a number of other pathways in addition to SPPL3. However, obtaining and evaluating the tissue to changes in CD19 glycosylation on relapsed leukemia and lymphoma samples will require development of novel tissue storage and technical assay pipelines. Few centers have robust biorepositories of relapsed leukemia or lymphoma samples, and thus making concrete conclusions is greatly limited by the availability of appropriate tissues. We have performed some preliminary studies

using available samples at the University of Pennsylvania and included these in **Reviewer Figure 1**. Please refer to our response to Reviewer 3 Comment 1 for a thorough explanation of these data.

Reviewers' Comments:

Reviewer #2:

Remarks to the Author:

I thank the authors for their extensive new work and data, which, in my opinion, has strengthened their manuscript substantially. I therefore, support this manuscript for publication.

Just one small remark regarding the rebuttal letter (comment 4), the loss PHA-L will not be due to overbranching, as PHA-L recognizes tetra-antennary N-glycans, the most branched N-glycan species. Thus, a loss of PHA-L actually reflects a loss of branching.

Reviewer #3:

Remarks to the Author:

In the revised manuscript, the authors did add some NEW data to highlight the clinical relevance of their findings. Now the paper was improved significantly and I recommend publishing it in NC.

Reviewer #5:

Remarks to the Author:

This reviewer appreciate the efforts of the authors. Unfortunately, in this reviewer' s opinion the clinical relevance of the observation remains weak.